# Sleep Deprivation Triggers Mitochondrial DNA Release in Microglia to Induce Neural Inflammation: Preventative Effect of Hydroxytyrosol Butyrate

**DOI:** 10.3390/antiox13070833

**Published:** 2024-07-12

**Authors:** Yachong Hu, Yongyao Wang, Yifang Wang, Yuxia Zhang, Zhen Wang, Xiaohong Xu, Tinghua Zhang, Tiantian Zhang, Shuangxi Zhang, Ranrui Hu, Le Shi, Xudong Wang, Jin Li, Hui Shen, Jiankang Liu, Mami Noda, Yunhua Peng, Jiangang Long

**Affiliations:** 1Center for Mitochondrial Biology and Medicine, The Key Laboratory of Biomedical Information Engineering of Ministry of Education, School of Life Science and Technology, Xi’an Jiaotong University, Xi’an 710049, China; huyachong@stu.xjtu.edu.cn (Y.H.); wangyysep@foxmail.com (Y.W.); yifang@stu.xjtu.edu.cn (Y.W.); yx_zhang@stu.xjtu.edu.cn (Y.Z.); wz15761635541@stu.xjtu.edu.cn (Z.W.); sanjie1993@stu.xjtu.edu.cn (T.Z.); anneshuang@stu.xjtu.edu.cn (S.Z.); huranruiuse@stu.xjtu.edu.cn (R.H.); leshi@xjtu.edu.cn (L.S.); xudongwang@xjtu.edu.cn (X.W.); jkliu@uor.edu.cn (J.L.); maminoda39@gmail.com (M.N.); 2School of Pharmacy, Chengdu Medical College, Chengdu 610500, China; xxh@cmc.edu.cn (X.X.); zhang_tinghua@cmc.edu.cn (T.Z.); 3State Key Laboratory of Toxicology and Medical Countermeasures, Beijing Institute of Pharmacology and Toxicology, Beijing 100850, China; jinli9802@163.com; 4Department of Nutrition and Food Hygiene, Faculty of Naval Medicine, Naval Medical University, Shanghai 200433, China; shenhui@smmu.edu.cn; 5School of Health and Life Science, University of Health and Rehabilitation Sciences, Qingdao 266071, China; 6Research and Educational Resource Center for Immunophenotyping, RUDN University, 6 Miklukho-Maklaya St, 117198 Moscow, Russia

**Keywords:** sleep deprivation, mtDNA release, oxidative stress, hydroxytyrosol butyrate, microglia, neural inflammation

## Abstract

Sleep deprivation (SD) triggers mitochondrial dysfunction and neural inflammation, leading to cognitive impairment and mental issues. However, the mechanism involving mitochondrial dysfunction and neural inflammation still remains unclear. Here, we report that SD rats exhibited multiple behavioral disorders, brain oxidative stress, and robust brain mitochondrial DNA (mtDNA) oxidation. In particular, SD activated microglia and microglial mtDNA efflux to the cytosol and provoked brain pro-inflammatory cytokines. We observed that the mtDNA efflux and pro-inflammatory cytokines significantly reduced with the suppression of the mtDNA oxidation. With the treatment of a novel mitochondrial nutrient, hydroxytyrosol butyrate (HTHB), the SD-induced behavioral disorders were significantly ameliorated while mtDNA oxidation, mtDNA release, and NF-κB activation were remarkably alleviated in both the rat brain and the N9 microglial cell line. Together, these results indicate that microglial mtDNA oxidation and the resultant release induced by SD mediate neural inflammation and HTHB prevents mtDNA oxidation and efflux, providing a potential treatment for SD-induced mental issues.

## 1. Introduction

Sleep is crucial for health [1]. Sleep deprivation (SD) causes multiple health problems, including mood disturbances, psychosis, neurological disorders, heart attack, hypertension, stroke, and cancer [2]. A critical consequence of SD is brain dysfunction [3,4], resulting in emotional and cognitive impairment [5]. The accumulation of oxidative stress and immune response are potential mechanisms of neural dysfunction induced by SD [6,7,8]. The microglial cells are highly responsive stress sensors in the brain and possess the ability to interact with neurons and the immune system [9]. Clinical research has shown that sleep behavior disorders lead to neural inflammation and dopaminergic dysfunction caused by microglial activation [10]. Among glial cells affected by SD [11,12,13], microglial activation plays a crucial role in neuroimmune responses [14,15]. SD activates the NF-κB pathway in the brain, increasing the levels of IL-6, TNF, and IL-1β [16,17].

Both animal models and clinical studies consistently report that SD elevates levels of reactive oxygen species (ROS) [4]. These observations suggest that clearing accumulated ROS may be a reason for the body’s need for sleep [18]. ROS cause damage to macromolecules like DNA, proteins, and lipids. Moreover, SD has shown a connection to elevated levels of 8-Oxo-2′-deoxyguanosine (8-oxo-dG), a marker of oxidative DNA damage [19]. Furthermore, ROS compromise the integrity of mtDNA, leading to the accumulation of mutations and damage [20]. SD also triggers protein oxidation and higher carbonylation levels within the cerebral cortex [21].

Mitochondria play a crucial role in the immune system through mtDNA releasing to the cytoplasm, extracellular space, or circulation, which activates innate immune responses, including NF-κB signaling, when cells are under stress. SD is a typical factor that triggers mitochondrial dysfunction [22,23]. In previous studies, SD of 72 h reduced mitochondrial respiratory function and altered cristae structure in rat brains [24,25]. Recent research has found that cytoplasmic mtDNA in aged microglia activates the cGAS-STING pathway, leading to neurotoxicity and the impairment of memory capacity [26,27]. Cell-free mtDNA serve as biomarkers of neural stress for astronauts who experience low sleep quality [28,29]. The Bak/Bax microchannel is one of the identified channels for mtDNA release [30]. Oxidized mtDNA is a key factor in its fragmentation and release [31]. However, the exact mechanism whereby SD induces neural inflammation driven by mtDNA release remains unclear.

Hydroxytyrosol (HT) is one of the functional polyphenolic compounds found in olive oil. Our previous study demonstrated that HT inhibited NF-κB signaling and relieved mitochondrial oxidative stress, which then reduced inflammation and oxidative stress [32]. Additionally, hydroxybutyric acid (HB) is a metabolite produced in the liver mitochondria that benefits brain cells by crossing the blood–brain barrier and providing energy [33]. Therefore, we synthesized a new compound named hydroxytyrosol butyrate (HTHB, also named HT-Bu, or 2-(3,4-dihydroxyphenyl)ethyl 3-hydroxybutanoate) [34]. When ingested, HTHB enters the bloodstream and is distributed to various major organs including the brain. It rapidly metabolizes into HT and HB without exhibiting toxicity [35,36]. We previously defined nutrients and natural compounds that stimulate mitochondrial metabolism as “mitochondrial nutrients” [37]. Therefore, HTHB is supposed to be a new potential mitochondrial nutrient that protects against neural damage by reducing mitochondrial oxidative stress and increasing cellular energy.

We identified that SD induces oxidative stress and mtDNA release, which are key mechanisms underlying neuroinflammation. Upon this, we found that HTHB exerts its anti-inflammatory effects by relieving oxidative-stress-induced mtDNA oxidation and the release to the cytosol in the cortex.

## 2. Materials and Methods

### 2.1. Materials and Antibodies

The HTHB molecule is a new mitochondrial nutrient designed and synthesized by our team [34]. We conducted preliminary verification of the safety and pharmacokinetics [35,36].

The following antibodies were used: Anti-TFAM (Santa Cruz, Dallas, TX, USA, sc-166965), Anti-GAPDH (Cell Signaling Technology, Boston, MA, USA, #5174), Anti-Iba1 (Cell Signaling Technology, Boston, MA, USA, #17198), Anti-CD68 (Cell Signaling Technology, Boston, MA, USA) Anti-NF-κB p65 (Cell Signaling Technology, Boston, MA, USA, # 8242), Anti-Phospho-NF-κB p65 (Ser536, Cell Signaling Technology, Boston, MA, USA, #3033), Anti-Actin (Cell Signaling Technology, Boston, MA, USA, #4967), Anti-Tubulin (Cell Signaling Technology, Boston, MA, USA, #5568), Anti-Histone (Cell Signaling Technology, Boston, MA, USA, #4499), Anti-Bak (Cell Signaling Technology, Boston, MA, USA, #12105), Anti-Bax (Cell Signaling Technology, Boston, MA, USA, #2772), Anti-VDAC (Cell Signaling Technology, Boston, MA, USA, #4661), and Anti-GFAP (Cell Signaling Technology, Boston, MA, USA, #12389).

### 2.2. Animals and Drug Treatments

Eight-week-old male rats were obtained from the Naval Medical University Animal Center (Shanghai, China). The rats were housed under a 12/12 h light/dark cycle, with free access to food and water throughout the experiment, and were kept at a temperature of 23–25 °C with natural humidity. After acclimatizing for one week, the rats were randomly divided into five groups: the control group (n = 15), the SD group (n = 10), the HT-treated SD group (SD + HT, n = 10), an HB-treated SD group (SD + HB, n = 10) and the HTHB-treated SD group (SD + HTHB, n = 10). The average body weights of each group were nearly identical (190–210 g). The rats were administered HT (35 mg/kg/day dissolved in water), HB (23.6 mg/kg/day dissolved in water), and HTHB (54.5 mg/kg/day dissolved in water) via oral gavage for 2 weeks. The molar concentrations of HT, HB, and HTHB were equal. The rats in the control and the SD groups without mitochondrial nutrients were administered the same volume of water via oral gavage for 2 weeks.

### 2.3. SD Model

SD is characterized by acute or chronic SD [38]. To mimic sleep disorders, in this study, we used the multiple-platform method, which deprives rats of almost all rapid eye movement (REM) sleep [39,40]. We selected 72 h of SD based on the previous experimental studies for more significant results as durations of 24 h [41,42], 48 h [43], and 72 h [44,45] are commonly used for acute SD.

The materials for the multiple-platform method for SD consisted of a water tank and a lid. The water tank (120 cm × 60 cm × 30 cm) contained two rows of ten circular platforms, each with a diameter of 6 cm. The platforms were connected to the bottom of the tank by iron rods measuring 8 cm in height, which elevated the platforms 8.0 cm above the bottom of the tank. The thickness of the platform and the tank wall was approximately 0.8 cm. Two rows of 10 small grooves were provided above the platforms to place water bottles and a large groove was provided between the two rows of small grooves to place feed. At the beginning of the experiment, the water tank was filled with water up to a level that was 1.0 cm below the surface of the platforms and the temperature of the water was maintained at 22 °C. The rats were placed on the platforms, the lid was closed, and sufficient water and feed were provided on the lid. After 72 h of SD, the rats were sacrificed. During the 72 h SD period, the rats also received corresponding drug treatment. In all experiments in this study, we focused on the primary somatosensory cortex of the brain.

The body weights were measured every 3 days and food intakes were recorded daily. After treatment with HT, HB, or HTHB for 11 days, all mice were subjected to the Morris water maze test (MWM) and the open field test (OFT). Rats were anesthetized using a sodium pentobarbital injection (40 mg/kg) before dissection. All of the experimental procedures followed the Guide for the Care and Use of Laboratory Animals, eighth edition (ISBN-10: 0-309-15396-4), and the animal protocol was approved by the Animal Use and Care Committee of Xi’an Jiaotong University (Approval No.: XJTU-2019-21).

### 2.4. Morris Water Maze Test

On the 11th day of the drug treatment, spatial acquisition experiments were performed on rats undergoing SD. Spatial acquisition lasted for 4 days, with the final day being a probe trial. The water maze was composed of a large circular black water pool with a diameter of 120 cm and a height of 50 cm, which was divided into four equally sized quadrants on the monitoring screen of a computer. Water was added until the water surface was 1.0 cm above the surface of the circular platform (diameter 8–15 cm), and non-toxic black ink was used for coloring. The water maze was placed in a laboratory with uniform and dim lighting, at a temperature of 26 ± 2 °C, and contained multiple obvious visual cues (black geometric shapes printed on A4 paper). The platform was ensured to be invisible below the water surface and the water was changed for each training session. The water temperature was maintained at 22 ± 2 °C. Each day, rats were placed in the first, second, and third quadrants in sequence for training. At the start of each trial in each quadrant, each rat was placed into the water facing one side of the pool. Once in the water, the rat was allowed to search for the escape platform for 120 s. If the rat did not find the platform within 120 s, the experimenter guided the rat to the platform. Once the rat climbed onto the platform, it was allowed to stay for 15 s to observe the spatial cues on the platform (if less than 15 s, it was guided again). The video tracking system was used to record the rat’s path before it discovered the hidden platform. During the four-day spatial acquisition period, it was ensured that the sequence of the quadrants in which the rats entered the water was randomized each day rather than consistent. During the probe trial, the platform was removed and each rat was placed into the fourth quadrant (the bottom right corner) to observe its movement trajectory, which lasted for 120 s. The water maze trajectory data were analyzed using JLBehv-MWMR 3.0 software (Shanghai Jiliang Software Technology, Shanghai, China).

### 2.5. Open Field Test

The open field test (OFT) was conducted in a box measuring 80 cm × 80 cm × 50 cm. At the beginning of the test, each rat was placed in one of the corners of the box and monitored by a camera placed above the box for a duration of 5 min. Following each trial, the device was cleaned with 75% alcohol to ensure cleanliness. The trajectory data were recorded using an automated analysis system equipped with a tracking system and analyzed with the JLBehv-LAR 3.0 software (Shanghai Jiliang Software Technology, Shanghai, China). The center of the open field apparatus, which covered approximately 25% of the total area, was marked out as a square measuring roughly 40 cm × 40 cm.

### 2.6. Enzyme-Linked Immunosorbent Assay (ELISA)

Isolating mitochondria from 20 mg cortical tissue or 10^6^ N9 cell line, we prepared mitochondrial homogenate, centrifuged it, and collected the supernatant. We used enzyme-linked immunosorbent assay (ELISA) kits (Abcam, Boston, MA, USA) to measure mitochondrial 8-oxo-dG levels in brain tissue. We prepared cortical or N9 cell homogenate, centrifuged it, and collected the supernatant. We used ELISA kits (Abcam, Boston, MA, USA) to measure 8-oxo-dG levels in brain tissue. The levels of 4-hydroxynonenal (4-HNE), malondialdehyde (MDA), TNF, IL-6, and IL-1β in serum were also quantified using ELISA kits (Abcam, Boston, MA, USA). The detection process was performed in accordance with the manufacturer’s guidelines and protocols.

### 2.7. ROS Detection

Fresh rat brain tissue (50 mg) was divided into two portions and homogenized using a homogenizer. The first portion was centrifuged to obtain the supernatant for measuring total ROS levels while the second portion underwent mitochondrial isolation through differential centrifugation to prepare a mitochondrial suspension for assessing mitochondrial ROS levels. Both samples were stained with DHE (Abcam, Boston, MA, USA) and fluorescence intensity was measured using a fluorescent microplate reader (Thermo Fisher Scientific, Waltham, MA, USA).

ROS levels in N9 microglial cells were labeled using Dichlorodihydrofluorescein diacetate (DCFH-DA) from ROS Assay Kits (Abcam, Boston, MA, USA). The method for detecting mitochondrial ROS in N9 microglial cells was similar to that for brain tissue. After homogenizing cellular samples, mitochondria were isolated and stained with DHE (Abcam, Boston, MA, USA). Fluorescence intensity was measured using a fluorescent microplate reader (Thermo Fisher Scientific, Waltham, MA, USA). We also utilized flow cytometry to measure mitochondrial ROS levels in N9 cells, using mitoSOX (1:1000, Invitrogen, Carlsbad, CA, USA) for cellular staining. Subsequently, after cell collection, a cytoFLEX (BECKMAN COULTER, Brea, CA, USA) flow cytometer was employed to detect fluorescent signals from 10,000 cells, followed by data analysis.

### 2.8. Immunohistochemistry (IHC)

Briefly, paraffin sections were dewaxed with water and subjected to antigen retrieval using sodium citrate buffer (pH 6.0). Endogenous peroxidase was blocked with 3% H_2_O_2_ in methanol. The sections were then incubated with 5% bovine serum albumin (BSA) for 1 h, followed by overnight incubation with primary antibody at 4 °C and then biotin-conjugated secondary antibody for 50 min after washing three times with phosphate-buffered saline (PBS). Horseradish peroxidase (HRP)-conjugated third antibody was added for another 50 min incubation followed by washing three times with PBS and then visualization with diaminobenzidine (DAB) solution. The slides were finally counterstained with Harris hematoxylin solution, dehydrated, and covered with neutral balsam. The anti-glial fibrillary acidic protein (GFAP) antibody (1:800) was used to stain astrocytes; anti-ionized calcium binding adaptor molecule 1 (Iba1) antibody (1:800) and anti-Cluster of Differentiation 68 (CD68) antibody (1:500) were used to stain microglia. The images were captured under a Zeiss Axioscope microscope (Oberkochen, Germany) and analyzed using Image J 154-64bit software.

### 2.9. Fluorescence IHC

The rat brain was subjected to fluorescence in situ hybridization (FISH) and immunofluorescence co-staining. Tissue was removed, washed, and fixed in fixing solution (4% paraformaldehyde) for 12 h. After fixation, the tissue was dehydrated in a graded series of alcohol and embedded in paraffin. The paraffin blocks were sectioned using a microtome, picked up with a slide warmer, and then baked at 62 °C for 2 h. The sections were washed in xylene, followed by dehydration in alcohol (100%, 85%, and 75%) and finally in diethylpyrocarbonate (DEPC)-treated water. The sections were boiled in the repair solution for 15 min and allowed to cool naturally. Proteinase K (20 μg/mL) was added and the sections were incubated for 30 min at 37 °C. After washing with pure water, the sections were washed three times for 5 min each with PBS. The sections were then incubated in pre-hybridization solution at 37 °C for 1 h, followed by incubation with hybridization solution containing the ND5 probe (Probe sequence: TTCATCGTTGAGGCTATGGAT, Qiagen, Valencia, CA, USA) for mtDNA overnight at 37 °C. The sections were washed with 2 × saline-sodium citrate (SSC) buffer at 37 °C for 10 min, followed by washes twice with 1 × SSC buffer at 37 °C for 5 min each and a final wash with 0.5 × SSC at room temperature for 10 min. The sections were then incubated with voltage-dependent anion channel (VDAC) primary antibody for mitochondrial labeling, diluted in PBS, at 4 °C overnight. After washing with PBS three times for 5 min each, the sections were incubated with the appropriate secondary antibody at room temperature for 50 min. The sections were washed again with PBS three times for 5 min each and then incubated with 4′,6-diamidino-2-phenylindole (DAPI) staining solution for 8 min in the dark. After washing, the sections were mounted with an anti-fluorescent quencher and observed and imaged under a Nikon upright fluorescent microscope (Tokyo, Japan).

Immunofluorescence staining was conducted on rat brain tissue sections. The sections were dewaxed in water and subjected to antigen retrieval with EDTA buffer (pH 8.0). The endogenous peroxidase was blocked using 3% H_2_O_2_. The slides were then incubated in BSA for 30 min, followed by incubation with primary antibodies overnight at 4 °C. After washing three times with PBS, the slides were incubated with HRP-conjugated secondary antibodies for 50 min. Subsequently, the slides were incubated with DAPI for 10 min, washed with PBS, and covered with a fluorescence quencher. Neurons were stained with anti-neuronal nuclei (NeuN) antibody (1:3000) while anti-tumor necrosis factor alpha (TNF) antibody (1:500) and anti-nuclear factor NF-kappa-B p65 (p65) (1:500) were used to stain TNF and NF-κB, respectively. The images were captured using a Zeiss LSM700 confocal microscope (Zeiss, Oberkochen, Germany) and analyzed with Image J 154-64bit software.

For N9 cell samples, cells were fixed with 4% paraformaldehyde (in DEPC water) for 20 min and washed three times with PBS (pH 7.4) on a decolorizing shaker for 5 min each time. Proteinase K (20 g/mL) was added and the cells were digested for 8 min. After washing with pure water, the cells were washed three times for 5 min each with PBS. The cells were then incubated in pre-hybridization solution at 37 °C for 1 h, followed by incubation with hybridization solution containing the mtDNA ND5 probe (Probe sequence: TGAAGTGAGGATAAGCATGGT, Qiagen, USA) overnight at 37 °C. The cells were washed with 2 × SSC at 37 °C for 10 min, followed by two washes with 1 × SSC at 37 °C for 5 min each and a final wash with 0.5 × SSC at 37 °C for 10 min. The cells were then incubated with the VDAC primary antibody for mitochondrial labeling, diluted in PBS, at 4 °C overnight. After washing with PBS three times for 5 min each, the cells were incubated with the appropriate secondary antibody at room temperature for 50 min. The cells were washed again with PBS three times for 5 min each and then incubated with DAPI staining solution for 8 min in the dark. After washing, the cells were mounted with an anti-fluorescent quencher and observed and imaged under a Nikon upright fluorescent microscope. All these antibodies were procured from Cell Signaling Technology, Boston, MA, USA.

### 2.10. Cell Culture

The N9 mouse microglial cells were obtained from the American Type Culture Collection (ATCC, Manassas, VA, USA) and cultured in DMEM supplemented with 10% fetal bovine serum (Thermo Fisher Scientific, Waltham, MA, USA) and 1% penicillin–streptomycin (Thermo Fisher Scientific, Waltham, MA, USA) at 37 °C in a 5% CO_2_ environment. N9 cells from passage 3–4 were used for the experiments.

### 2.11. Detection of mtDNA Content in Cytosolic Extracts

Equal aliquots of cultured N9 microglial cells (10^6^) or fresh rat brain tissue (50 mg) were resuspended in either DNA extraction buffer or buffer with digitonin. The former served as controls for total mtDNA while the latter was incubated and centrifuged to obtain pure cytosolic fractions. DNA was isolated from the fractions using a Rapid Animal Genomic DNA Isolation Kit (Beyotime, Shanghai, China) and quantitative PCR was performed using both nuclear and mtDNA primers. The cycle threshold values obtained for mtDNA abundance from whole-cell extracts were used as normalization controls for the mtDNA values obtained from cytosolic fractions. Appendix A provides a list of DNA primers used in this study [46].

### 2.12. RNA Isolation and Real-Time PCR

According to the manufacturer’s protocol, total RNA was extracted from 10 mg of brain tissue using TRI reagent (Sigma-Aldrich, Darmstadt, Germany) and reverse-transcribed into cDNA using PrimeScriptTM RT Master Mix (TaKaRa Bio, Shiga, Japan). Real-time PCR was performed with different primers using a TB Green Premix Ex Taq II (TaKaRa Bio, Shiga, Japan) reagent kit. Normalization of mRNA levels was achieved by comparing them to β-actin mRNA levels. The list of RNA primers used in this study is shown in Appendix A.

### 2.13. Western Blot

The brain tissue was homogenized in cell lysis buffer (Abcam, Boston, MA, USA) supplemented with PMSF for Western blotting and immunoprecipitation (IP). The lysate was centrifuged at 13,000× *g* for 15 min to collect the supernatant, and the protein concentration was measured using a PierceTM BCA Protein Assay Kit (Thermo Fisher Scientific, Waltham, MA, USA). Next, the protein samples were separated by 10% sodium dodecyl sulfate–polyacrylamide gel electrophoresis (SDS-PAGE) and transferred onto nitrocellulose membranes. After blocking with 5% nonfat milk in Tris-buffered saline Tween (TBST) for 1 h, the membranes were incubated with primary antibodies (1:1000) overnight at 4 °C. The membranes were then washed with TBST and incubated with secondary antibodies (1:2000) for 1 h at room temperature. The membranes were washed again with TBST and detected by chemiluminescence using Clarity Western ECL Substrate (Bio-Rad Laboratories, Hercules, CA, USA) or Immobilon Western Chemiluminescent HRP Substrate (Millipore, Darmstadt, Germany) reagents. The signals were analyzed using Image J 154-64bit software.

### 2.14. Transmission Electron Microscope (TEM)

TEM experiments were conducted at the Electron Microscopy Laboratory at the Health Science Center (HSC) of Xi’an Jiaotong University. Brain tissues (approximately 1 cubic millimeter) were flat-fixed in 1% osmium tetroxide, dehydrated in ascending ethanol concentrations, treated with propylene oxide, and then impregnated with resin overnight at room temperature. The embedded tissues were mounted between ACLAR embedding films and cured at 55 °C for 72 h. Areas of interest were excised from the embedding films, re-embedded at the tip of resin blocks, and cut into 65–80 nm thick sections using an ultramicrotome (Leica Ultracut UC7, Leica, Wetzlar, Germany). The sections were collected on copper square mesh grids, and at least 30 mitochondria from at least 3 different rats were randomly imaged at a magnification of 30,000× for each sample using an H-7650 (Hitachi, Tokyo, Japan) transmission electron microscope at 80 kV. The morphology of the mitochondria was manually quantified.

### 2.15. Statistical Analysis

Statistical analyses were performed using GraphPad Prism 9 (GraphPad Software, Boston, MA, USA). One-way analysis of variance (ANOVA), two-way ANOVA, or Student’s *t*-test was utilized to determine differences among groups. The data were expressed as means ± standard errors of the mean (SEMs). All data underwent Kolmogorov–Smirnov normality test. Significance was considered as * *p* < 0.05, ** *p* < 0.01, or *** *p* < 0.001.

## 3. Results

### 3.1. The Declines in Learning and Memory Abilities and Emotional Deficits in Sleep-Deprived Rats and the Alleviation by HTHB

To create an animal model of SD, 10-week-old rats were subjected to 72 h of SD, followed by the Morris water maze (MWM) and open-field test (OFT) as scheduled in Figure 1A. SD caused significant weight loss, irritability, and increased aggression, which was consistent with previous research [3]. Both the SD group and the other (SD + HT, SD + HB, or SD + HTHB) groups showed a 10% decrease in body weight (Appendix A) and increased food consumption (Appendix A). The increase in food consumption induced by SD was not solely due to changes in food intake. We observed that the water inside the experimental tank contained large amounts of food debris, suggesting that the sleep-deprived rats were constantly gnawing at their food instead of swallowing it, possibly due to abnormal moods.

To evaluate spatial learning and memory abilities, we performed the MWM test and found that the SD group showed a longer escape latency for spatial acquisition compared to the control group on the fourth day (Figure 1B). The representative search paths in the probe trial test of the MWM are shown in Figure 1C. SD rats swam at a faster speed (Figure 1D), spent less time (Figure 1E), and swam a shorter distance (Figure 1F) in the quadrant of the probe (IV quadrant) than the control group during the probe trial of the MWM. To assess mood changes, the OFT was used. The representative moving paths in the OFT are shown in Figure 1G. SD rats had a longer moving distance in the central region, with faster speed, and increased frequency (Figure 1H–J). These results suggest that SD brings significant changes in mood and cognitive function.

To assess the efficacy of the compounds, HT, HB, and the newly synthesized HTHB, all mitochondrial nutrients, were compared in behavioral tests. Compared with HT and HB, HTHB significantly reduced the latency time in the 4th spatial acquisition and improved the distance/time ratio in the probe-quadrant (IV quadrant) of the MWM (Figure 1E,F). In the OFT, HT and HTHB reduced the SD-induced ratio of inner/outer field distance (Figure 1H). Additionally, HT and HTHB attenuated the SD-increased average speed (Figure 1I). On the other hand, HT and HTHB reversed the SD-induced decrease in the movement number (Figure 1J). These results indicate that during SD, HTHB exerts more pronounced effects on learning and memory, extending the improvement in emotional function observed with HT and HB.

### 3.2. Mitochondrial Oxidative Stress and mtDNA Release in Sleep-Deprived Rats and Improvement of Mitochondrial Function by HTHB

We further explored the effects of the mitochondrial nutrient HTHB on oxidative stress and mitochondrial function. SD led to higher levels of ROS and 8-oxo-dG in both brain tissue mitochondria (Figure 2A–D) and serum (Appendix A), indicating significant mitochondrial oxidative stress. However, HTHB attenuated SD-induced mitochondrial ROS and 8-oxo-dG levels (Figure 2A–D). The levels of 4-hydroxynonenal (4-HNE) and malondialdehyde (MDA) in the serum were also significantly increased in the SD group; however, HTHB attenuated them (Appendix A).

The release of mtDNA into the cytosol is considered a major trigger of the immune response [47]. To confirm mtDNA release in sleep-deprived rats, we observed an increase in cellular and cytoplasmic mtDNA levels under SD conditions (Figure 2E,F). We also detected a higher expression of mitochondrial transcription factor A (TFAM) in the SD group than in the control group (Figure 2G,H). TFAM regulated transcription, replication, and packaging of mtDNA. Recent studies have found that TFAM binds to mtDNA and triggers mtDNA release from BAK/BAX micropores [48].

Transmission electron microscopy (TEM) images of brain mitochondria (Figure 2I) revealed a reduction in the mitochondrial cross-sectional area, suggesting a shrinkage of mitochondria (Figure 2J), as well as a decrease in cristae number and density (Figure 2K,L), and increased mitochondrial vacuolization (Figure 2M) in the SD group compared to the control group. HTHB showed significant improvement in SD-induced reductions in the mitochondrial cross-sectional area and mitochondrial cristae number (Figure 2J,K).

These studies demonstrated that SD induced mitochondrial oxidative stress and mtDNA release, which affected mitochondrial structure. HTHB inhibited mtDNA release and protected mitochondrial structure and function.

### 3.3. Brain Microglial Activation, Mitochondrial DNA Release, and Neuronal Loss in SD and Alleviation by HTHB

Since microglial cells constitute the main immune cell population in the brain, we checked whether microglial cells were activated by SD. Immunohistochemical staining using CD68 and Iba1, markers of activated microglia/macrophages and microglia/macrophage-specific proteins, respectively, was performed (Figure 3A,C). The densities of CD68- and Iba1-positive cells in the SD group were significantly higher than those in the control group (Figure 3B,D), indicating an increase in the number and activation of microglia in the brains of SD rats [49]. Consistent with this finding, Western blotting showed a higher protein expression of Iba1 in the SD group compared to the control group (Figure 3E,F). HTHB attenuated SD-induced densities of CD68- and Iba1-positive cells and expression levels of Iba1, suggesting that HTHB inhibited microglial activation (Figure 3A–F). We analyzed the morphology of microglial branches in Appendix A. SD decreased both the complexity (S7B) and total length (S7C) of microglial branches while HTHB mitigated these effects. To further investigate, we used Iba1 labeling to identify microglial cells together with ND5 probes to label mtDNA and mitochondrial import receptor subunit TOM20 to label mitochondria (Figure 3G). It is noteworthy that following SD, the co-staining of TOM20 and ND5 in microglial cells significantly decreased, indicating the leakage of mtDNA from mitochondria (Figure 3H). Notably, mtDNA leakage was significantly lower in the SD + HTHB group compared to the SD group (Figure 3G,H). On the other hand, the immunohistochemical positive area and protein expression levels of GFAP in the SD group were not significantly different from those in the control group (Appendix A), indicating that astrocytes were not activated in the brains of SD rats, although they have been reported to play an important role in SD [12,50]. These results suggested that SD activated microglia, leading to microglial mtDNA release and resulting in neuronal loss. HTHB appeared to have a protective effect against these consequences of SD.

We also investigated the effects of SD on neuronal function in rats. NeuN immunofluorescence density was lower in the SD group, indicating neuronal loss in the cerebral cortex (Appendix A). However, HTHB rescued SD-induced neuronal loss (Appendix A). We detected mRNA levels of multiple cytokines related to neuronal function in rat brain tissue. The N-methyl-D-aspartate receptor (NMDAR) is an important excitatory glutamate receptor; however, the abnormal overexpression of NMDAR can lead to mitochondrial dysfunction, excessive glutamate response, excitotoxicity, and neuronal death [51,52]. The upregulation of NMDAR levels suggested that brain cells may have experienced excitotoxicity (Appendix A). The downregulation of VGF gene (non-acronymic) and nerve growth factor (NGF) levels suggested that SD may trigger an inflammatory response, affecting the metabolism and survival of neurons, which could be related to emotional deficits (Appendix A). HTHB inhibited the loss of neurons in the cerebral cortex and altered the levels of NMDAR and NGF in sleep-deprived rats (Appendix A).

### 3.4. Attenuation of NF-κB p65 Activation and Inflammation by HTHB

NF-κB is a critical pathway implicated in the activation of neural inflammation induced by SD [6]. As previously reported [45], microglial activation in the brains of SD rats was accompanied by NF-κB activation. After SD, the level of NF-κB increased in both the nucleus and cytoplasm of cerebral cortex cells, as demonstrated in our study as well (Figure 4A–G). Results from both immunofluorescence staining (Figure 4A–C) and Western blot (Figure 4D–G) revealed that HTHB significantly reduced the activation of NF-κB in the cortex, particularly NF-κB in the nucleus (Figure 4G). The activation of NF-κB in SD rats was associated with various inflammatory factors including TNF and IL-6. TNF immunofluorescence staining on brain tissue sections (Figure 4H) showed that SD rats showed higher fluorescence density compared to the control group, which was attenuated by HTHB (Figure 4I). The mRNA levels of TNF and IL-6 in SD rats were significantly higher than those in the control, which were attenuated by HTHB as well (Figure 4J,K). We further checked the serum levels of TNF, IL-6, and IL-1β. The SD-induced increase in the concentration of those pro-inflammatory cytokines in the serum was significantly attenuated by HTHB (Figure 4L–M).

### 3.5. Mitochondrial Oxidative Stress and mtDNA Release in LPS/ATP Induced Inflammatory N9 Cells Alleviated by HTHB

We then applied 1 μg/mL LPS and 1 mM ATP to cultured N9 microglia to mimic SD-induced microglia inflammation. ATP treatment promoted mitochondrial stress and induced mtDNA release [31]. The levels of mitochondrial ROS, as indicated by Dihydroethidium (DHE) staining, and mitochondrial 8-oxo-dG were significantly increased in the LPS/ATP group (Figure 5A,B and Appendix A). Using MitoSOX staining, we measured the mitochondrial ROS levels by flow cytometry and found that LPS/ATP induced cellular mitochondrial ROS levels (Figure 5C,D). Additionally, we used fluorescence in situ hybridization to stain mtDNA (mitoND5, green) and immunofluorescence to stain mitochondria (VDAC, red) and calculated the Pearson correlation coefficient of green and red pixels in each group (Figure 5E). The control group showed a higher degree of overlap between mtDNA and mitochondria while the LPS/ATP group showed mtDNA efflux out of mitochondria, as seen in the scatter plot of red and green fluorescence (Figure 5F). LPS/ATP treatments promoted the release of mtDNA (Figure 5G). Furthermore, LPS/ATP promoted NF-κB activation, resulting in a significant increase in both phosphorylated NF-κB levels and NF-κB expression (Figure 5H–J). The activation of p65 was accompanied by an elevation in the transcription levels of the inflammatory factors IL-6 and TNF (Figure 5K,L). These results demonstrate that LPS/ATP insult promotes mitochondrial oxidative damage, inducing mtDNA release and inflammation.

To confirm that mitochondrial oxidative stress had occurred prior to mtDNA release and inflammation, and to verify the role of HTHB, we investigated the effect of the antioxidant N-acetyl cysteine (NAC, 0.5 mM) and mitochondrial nutrient HTHB (10 μM), and both compounds reduced mitochondrial and cellular ROS levels (Figure 5A and Appendix A). HTHB or NAC also reduced both mitochondrial and cellular 8-oxo-dG levels (Figure 5B and Appendix A) as well as MitoSOX densities (Figure 5C,D). The degree of overlap between mtDNA and mitochondria in the LPS/ATP + HTHB and LPS/ATP + NAC groups was significantly higher than in the LPS/ATP group (Figure 5E,F). HTHB and NAC treatments inhibited cytoplasmic mtDNA levels (Figure 5G). Notably, both HTHB and NAC exhibited the ability to suppress p65 activation (Figure 5H–J), the transcription of IL-6 and TNF (Figure 5K,L), and improved cell viability damaged by LPS/ATP (Appendix A). These results clarify the critical role of mitochondrial oxidative stress in mtDNA efflux and the resultant inflammation.

### 3.6. Requirement of Bak for mtDNA Release but No Effect of Using HTHB

To explore whether the mitochondrial Bak/Bax channel is involved in mtDNA release, we found a higher expression of Bak, but not Bax and VDAC, in the brain tissue in SD rats (Figure 6A,B and Appendix A). We selected three segments from the D-loop and one segment from ND4 to assess the levels of mtDNA in the cytoplasm in N9 cells [46]. The knockdown of Bak alleviated the increase in cytoplasmic mtDNA (D-loop and ND4) levels induced by LPS/ATP (Figure 6C–E). The knockdown of Bak both significantly inhibited the activation of NF-κB (phosphorylated NF-κB) (Figure 6F,G) as well as suppressed the expressions of TNFa-mRNA and IL6-mRNA (Figure 6H,I). These results suggest that mtDNA release is more likely to occur via the Bak/Bax micropore rather than the VDAC oligomer. HTHB treatment significantly inhibited LPS/ATP-induced cytoplasmic mtDNA levels, the activation of NF-κB, and the expression of inflammatory cytokine mRNAs, with little effect on BAK expression (Figure 6E–I).

## 4. Discussion

Our study investigated the underlying mechanisms of SD-induced inflammatory responses and the protective effects of HTHB. We identified mitochondrial oxidative stress and microglial mtDNA release in the brain in SD rats and demonstrated that mtDNA release bridges mitochondrial oxidative stress and brain inflammation during SD (Figure 7).

Sleep disorders have been studied from various aspects including circadian rhythm disturbance, abnormal glucose metabolism, and sleep–immune imbalance. SD promotes systemic inflammation, including in the nervous system. The continuous activation of the inflammatory response by SD is a factor in the pathogenesis of neurodegenerative disorders [9]. During non-pathogen infections, SD interrupts inflammatory homeostasis by acting on inflammatory cytokines such as IFN, IL-6, and IL-1β [6]. SD activates microglia significantly [53,54,55], which plays a crucial role in immune responses [14,15]. Compelling evidence supports the notion that microglial activation is a fundamental factor contributing to the decline in learning and memory abilities observed after SD [56,57]. A recent study used LPS/ATP insult to mimic the stress induced by SD in N9 microglia cells [49]. Some studies have shown that SD activates both astrocytes and microglia in the brain, leading to inflammation cytokine and nerve damage [14,17]. However, we demonstrated that 3-day SD induced microglia activation rather than astrocytes in the study (Appendix A), suggesting that microglial cells are more susceptible to SD in the early phase. The duration and intensity of sleep deprivation are crucial for activating different types of glial cells. In 72 h continuous sleep deprivation, microglia are more likely to be activated than astrocytes [15].

Mitochondria play a critical role in inducing exogenous stress that leads to neural inflammation, with the release of mtDNA into the cytosol, which is considered a major trigger of the immune response [47,58]. Oxidative damage to mitochondria, including damage to mtDNA, is considered a source of the inflammatory response [7]. Studies have shown that SD induces mitochondrial oxidative stress [4]. We have provided the first evidence that SD provokes neural inflammation via microglial mtDNA release. Although some mitochondrial-related experiments were conducted in purified mitochondria or using mitochondrial-targeted dyes, the mitochondrial-specific data remained insufficient, which was the major limitation of this study.

Studies have shown that oxidative-damaged mtDNA can escape from mitochondria through channels such as the Bak/Bax and VDAC oligomer channels [30,31,59,60]. Bak/Bax macropores are more likely to be activated under higher stress than oligomerized VDAC [59]. However, the role of the mitochondrial permeability transition pore (mPTP) is still unclear [30,31,59,61]. Our data suggest that mtDNA effluxes through Bak/Bax micropores instead of VDAC oligomers during SD (Appendix A). Under stress conditions, intracellular mtDNA release triggers the inflammatory response through the TLR9, cGAS-STING, and NLRP3 pathways, which are associated with the activation of the inflammatory factor NF-κB and can further activate other inflammatory factors (IL-6 and TNF) [31,46,62]. Interestingly, we observed that sleep deprivation led to an overall increase in cortical mtDNA levels (Figure 2E), possibly indicating the compensatory synthesis of mtDNA following its damage.

Our previous studies have shown that both HT and HB cross the blood–brain barrier and improve neurological function in AD mice by inhibiting NF-κB activation, and HT also has the effect of reducing mitochondrial oxidative stress [32,33]. HTHB is a mitochondrial nutrient that was designed and synthesized in our laboratory [34]. It showed no toxicity in mice and is supposed to have synergistic effects of antioxidative stress and anti-inflammation [35,36]. In the present study, compared with the HT or HB treatment, the new compound HTHB exhibited better behavioral performance under SD conditions, suggesting possible synergistic effects of HTHB. Considering the importance of mitochondrial function in multiple systems of the body, we hypothesize that the mitochondrial nutrient HTHB may mitigate metabolic abnormalities, cardiovascular damage, and immune impairment induced by SD. However, more pharmacokinetic data should be measured to explore the detailed mechanism of HTHB’s impact.

This study primarily focused on the primary somatosensory cortex of the brain, which is associated with both emotions [63] as well as learning and memory abilities [64]. The multiple-platform method is a common approach for SD, but it may not eliminate other types of stressors. These stressors could potentially impact microglial cells as well. SD has an impact on synaptic plasticity, which, in turn, affects learning and memory [65,66,67]. Additionally, it increases the risk of forming inaccurate memories [68]. Both clinical and animal studies have demonstrated a connection between SD and emotional disorders [69], as well as an increase in aggression and violent behavior [70,71]. Sleep-deprived mice have exhibited manic behaviors, including excessive locomotion [42]. In the behavioral tests, we found that rats were more likely to move in the internal area of the open field, which represents the abnormal behavior of their emotions. However, other studies have found that SD rats are less active in the inner regions of the open field, which is considered a sign of depression [40,72]. This difference is not fundamentally contradictory since depression and mania alternate in a cycle known as bipolar disorder. Distinctions in the length, manner, and duration of SD may lead to different emotional states [73,74]. Considering the OFT as one of the experimental methods to examine animal sociability [75], we hypothesize that sleep deprivation and HTHB may influence social behavior in rats. Previous studies have shown that acute sleep deprivation in humans leads to reduced willingness to interact with others.

## 5. Conclusions

In conclusion, our study provided evidence that SD stimulates oxidative stress to induce microglial mtDNA release to the cytosol, triggering inflammatory responses and neuronal loss in rat brains. The study shed light on the understanding of the mechanisms of sleep-deprivation-involved neural immune response.

## Figures and Tables

**Figure 1 antioxidants-13-00833-f001:**
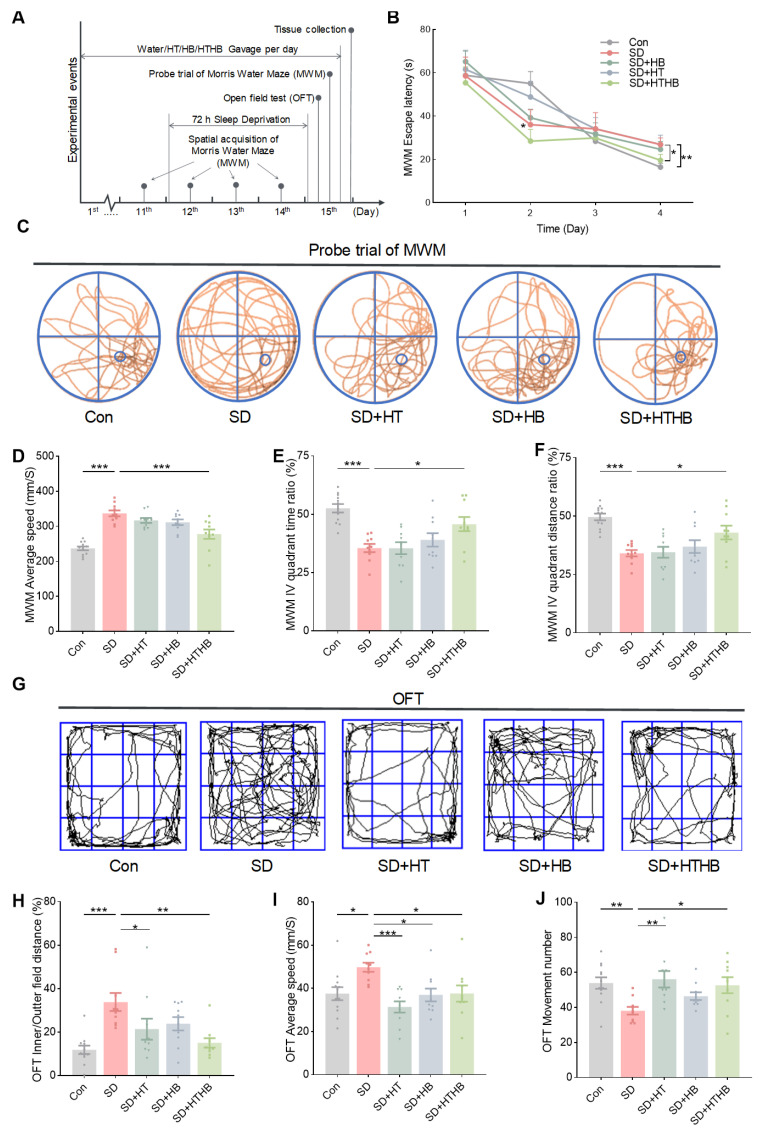
Morris water maze (MWM) test and open field test (OFT) in the sleep-deprived rat model and the comparative effects of hydroxytyrosol (HT), β hydroxybutyric acid (HB), and hydroxytyrosol butyrate (HTHB). (**A**) Schematic design of the sleep deprivation procedure and behavior tests for MWM and OFT. (**B**) Escape latency during the spatial acquisition phase in MWM for 4 days (days 11 to 14). (**C**) Representative tracking plot during the probe trial test in the MWM. (**D**) Average swim speed of the probe trial test in the MWM. (**E**,**F**) Time ratio (**E**) and distance ratio (**F**) in the target quadrant of the probe trial test in the MWM. (**G**) Representative tracking plot of the OFT. (**H**) Inner/outer field distance ratio, (**I**) average speed, and (**J**) the movement number in the OFT. Values are means ± SEMs, * *p* < 0.05, ** *p* < 0.01, *** *p* < 0.001. Con, n = 12; SD, n = 10; SD + HT, n = 10; SD + HB, n = 10; SD + HTHB, n = 10. The results were analyzed using two-way ANOVA.

**Figure 2 antioxidants-13-00833-f002:**
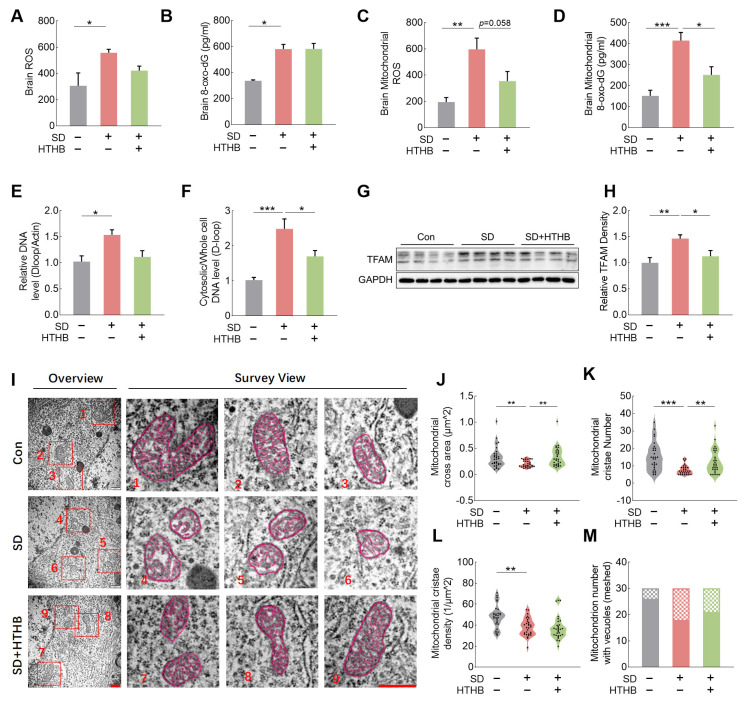
Sleep deprivation (SD) induces brain mitochondrial oxidative stress, morphology damage, and elevated cytoplasmic mtDNA levels, which are ameliorated by HTHB. (**A**,**B**) The levels of reactive oxygen species (ROS) (**A**) and 8-oxo-dG (**B**) in the rat brain, n = 4. (**C**,**D**) Mitochondrial levels of ROS (**C**) and 8-oxo-dG (**D**), n = 4. (**E**) Relative expression of cellular mtDNA, n = 8. (**F**) Relative ratio of cytosolic/whole-cell expression of mtDNA, n = 8. (**G**,**H**) Representative Western blot images of mitochondrial transcription factor A (TFAM) (**G**) and relative expression densities of TFAM (**H**), n = 8. (**I**) Transmission electron microscopy (TEM) imaging of mitochondria in the cerebral cortex in control, n = 30. Scale = 500 nm. (**J**) Analysis of mitochondrial cross-sectional area. (**K**) Analysis of cristae number per mitochondrion. (**L**) Analysis of mitochondrial cristae density (cristae number/1 mm^2^). (**M**) Number of mitochondria with vacuoles (meshed) and number of mitochondria with no vacuoles (non-meshed) out of all 30 mitochondria. Values are means ± SEMs, n ≥ 3, * *p* < 0.05, ** *p* < 0.01, *** *p* < 0.001. The results were analyzed using one-way ANOVA.

**Figure 3 antioxidants-13-00833-f003:**
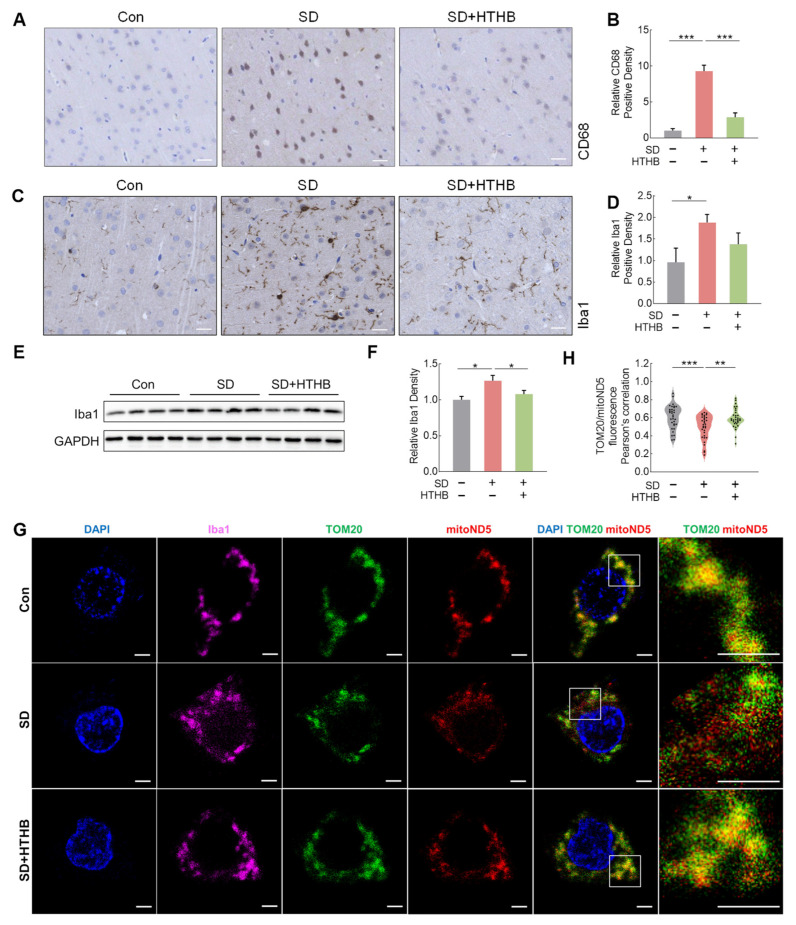
SD activates microglia and promotes mtDNA leaking into cytosol, which are ameliorated by HTHB. (**A**,**B**) Immunohistochemistry of CD68 (**A**) in a rat brain slice and the relative expression quantification of CD68-positive areas (**B**). Scale = 25 mm, n = 4. (**C**,**D**) Immunohistochemistry of Iba1 (**C**) in rat brain slices and quantification of positive areas (**D**), n = 4. (**E**,**F**) Western blot of Iba1 in rat cortex (**E**) and the relative expression density of Iba1 (**F**), n = 8. (**G**) Co-staining of DAPI (blue), Iba1 (pink), an important receptor for many mitochondrial pre-proteins (TOM20, blue), and mtDNA (mitoND5, red). The white box indicates the area that is magnified (on the right). Scale bar = 2 μm, n = 30 (from 3 rat brain slices). (**H**) mitoND5/TOM20 Pearson’s Colocalization Coefficients. Values are means ± SEMs, n ≥ 3, * *p* < 0.05, ** *p* < 0.01, *** *p* < 0.001. The results were analyzed using one-way ANOVA.

**Figure 4 antioxidants-13-00833-f004:**
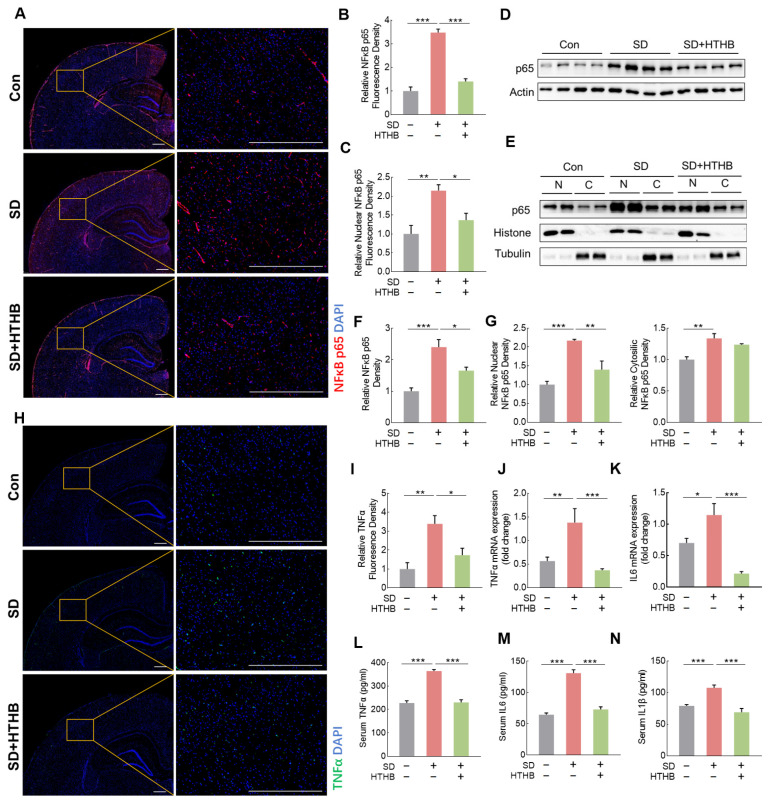
HTHB reduces brain inflammation in sleep-deprived rats. (**A**) Immunofluorescence imaging of NF-κB p65 (red) and nuclear (DAPI, blue) in the cortex area of the brain. Scale = 500 μm, n = 4. (**B**) Quantification of fluorescence density of the NF-κB p65-positive area. (**C**) Quantification of the fluorescence density of NF-κB p65 and DAPI (blue) co-localization area, n = 4. (**D**) Representative Western blot images of NF-κB p65 in brain tissue in rats. Quantification of NF-κB p65 protein expression levels, n = 4. (**E**) Representative Western blot images of NF-κB p65, histone, and tubulin in the nucleus and cytoplasm of the brain. (**F**) Relative expression density of NF-κB p65, n = 4. (**G**) Relative expression densities of nuclear and cytosolic NF-κB p65, n = 4. (**H**) Immunofluorescence staining of TNF (green) in the cortex area of the brain. Scale = 500 mm, n = 4. (**I**) Relative fluorescence intensity of TNF-positive area. (**J**,**K**) mRNA levels of TNF (**J**) and IL6 (**K**), n = 4. (**L**,**M**,**N**) Serum levels of TNF (**L**), IL-6 (**M**) and IL-1β (**N**), n = 8. Values are means ± SEMs, n ≥ 3, * *p* < 0.05, ** *p* < 0.01, *** *p* < 0.001. The results were analyzed using one-way ANOVA.

**Figure 5 antioxidants-13-00833-f005:**
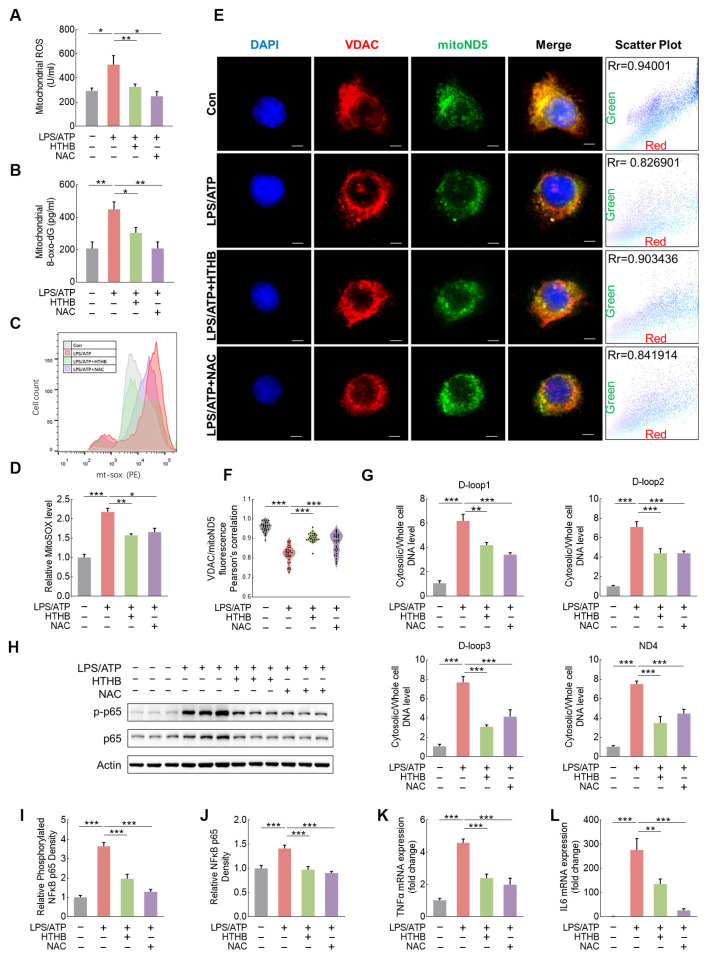
Mitochondrial oxidative stress and mtDNA release in LPS/ATP-induced inflammatory N9 microglial cells are ameliorated by HTHB. (**A**,**B**) ROS (**A**) and 8-oxo-dG (**B**) levels in mitochondria of N9 microglial cells, n = 4. (**C**,**D**) Flow cytometry (**C**), and statistical analysis (**D**) of MitoSOX staining, n = 3. (**E**) Co-staining of outer mitochondrial membrane protein VDAC (red) and mtDNA (mitoND5, green). Scale = 2 μm, n = 28. (**F**) Statistical analysis of co-localization of VDAC/mitoND5. (**G**) Ratio of cytoplasmic to cellular mtDNA expression. Loop1/Loop2/Loop3 is the sequence from mtDNA D-loop; ND4 is the sequence from mtDNA ND4, n = 4. (**H**) Representative Western blot images of phosphorylated NF-κB p65 (p-p65) and NF-κB p65 (p65) in N9 microglial cells. (**I**,**J**) Relative expression densities of p-NF-κB p65 (**I**) and NF-κB p65 (**J**), n = 6. (**K**,**L**) mRNA levels of TNF (**K**) and IL6 (**L**), n = 4. Values are means ± SEMs, n ≥ 3, * *p* < 0.05, ** *p* < 0.01, *** *p* < 0.001. The results were analyzed using one-way ANOVA.

**Figure 6 antioxidants-13-00833-f006:**
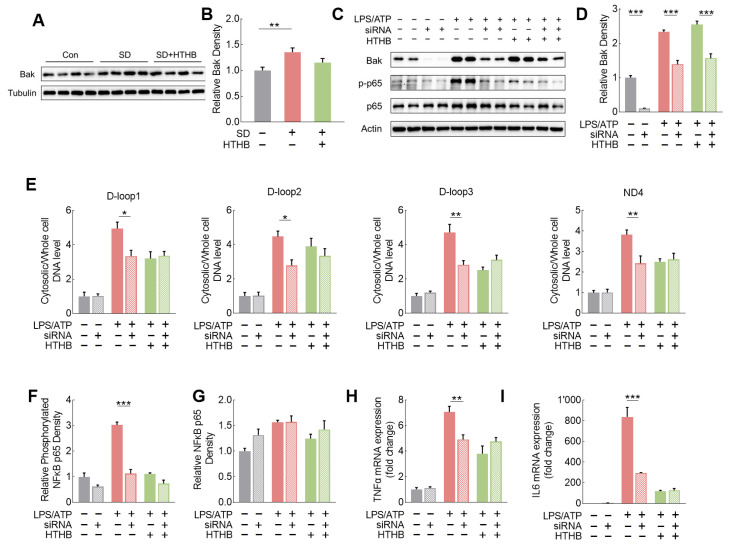
Upregulation of Bak is involved in mtDNA release but not impacted by HTHB. (**A**) Representative Western blot images of Bak in the cortex. (**B**) Relative expression density of Bak in the cortex, n = 8. (**C**) Representative Western blot images of Bak, phosphorylated NF-κB p65 (p-p65) and NF-κB p65 (p65), and the effects of siRNA and HTHB in N9 microglial cells. (**D**) Relative expression density of Bak with or without siRNA and HTHB, n = 4. (**E**) Ratio of cytoplasmic to whole-cell mtDNA levels. Loop1/Loop2/Loop3 is sequence from mtDNA D-loop, n = 4. ND4 is sequence from mtDNA ND4. (**F**,**G**) Relative expression densities of phosphorylated NF-κB p65 (**F**) and NF-κB p65 (**G**), n = 4. (**H**,**I**) Relative expression levels of mRNA of TNF (**H**) and IL6 (**I**), n = 4. Values are means ± SEMs, n ≥ 3, * *p* < 0.05, ** *p* < 0.01, *** *p* < 0.001. LPS: Lipopolysaccharide; ATP: adenosine triphosphate. The results were analyzed using one-way ANOVA, two-way ANOVA, or Student’s *t*-test.

**Figure 7 antioxidants-13-00833-f007:**
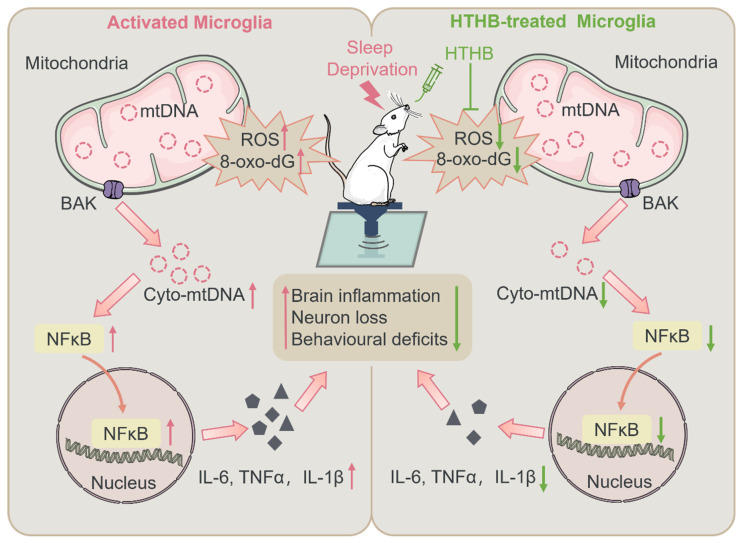
Sleep deprivation triggers mitochondrial DNA release in microglia to induce neural inflammation, which is prevented by hydroxytyrosol butyrate. SD promotes ROS and 8-oxo-dG levels in rat brain. Mitochondrial oxidative stress is the driver of mtDNA release. During SD, microglial mtDNA release activates NF-κB p65 activation and inflammation. HTHB effectively curbs mtDNA release by mitigating mitochondrial oxidative damage and alleviates brain inflammation, neuron loss, and behavioral deficits. The upward arrow indicates upregulation, and the downward arrow indicates downregulation. Red represents the effects induced by SD, while green represents the effects induced by HTHB.

## Data Availability

Data will be made available on request.

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
