# Peer review of "Sleep Deprivation Triggers Mitochondrial DNA Release in Microglia to Induce Neural Inflammation: Preventative Effect of Hydroxytyrosol Butyrate"

_antioxidants, 2024, doi:10.3390/antiox13070833_

Round 1

Reviewer 1 Report

This study shows that sleep-deprivation (SD) induces significant change in mood and cognitive function which might be induced by the induction of oxidative stress, the release of mitDNA and the brain inflammation involved in microglial cells in rats. The story seems theoretical, the methods used are appropriate and the results are clear. I have some minor concerns to realize this study.  

1. L107: “three” must be “five”.

2.L132: “dried” may be “sacrificed”.

3.Around L414: Although astrocytes were not changed in this study, some other studies indicated the involvement of astrocytes in SD as the authors mentioned. What are the differences among those studies? The simple explanation should be drawn here.

4.L456: In Fig. 5E the green signal is mitoND5 for mitDNA and the red one is outer mitochondrial membrane protein VDAC for mitochondria. These signals should be drawn also in the text “Results”. 

5. L460: What is the stimulation of co-treatment of LPS and ATP on N9 cells? The reason of the addition of ATP should be written here. Furthermore the explanation that the use of co-treatment could indicate the models of SD in N9 cells should be drawn in the text.

6. In Fig.6E, these figures have the words “D-loop1,2,3” but there are no explanation in the text. The authors should explain the reason to test in “D-loop1,2,3” and the results gained from these data in the lines 479-487 in the text “Results”. 

Reviewer 2 Report

Hu et al. have explored the effect of hydroxytyrosol butyrate on the effects of sleep deprivation. The results indicated that sleep deprivation induced oxidative stress and damage to the structure of the mitochondria. In addition, sleep deprivation activated microglia and caused neuronal inflammation. Treatment with a novel nutrient, hydroxytyrosol butyrate, these disorders were significantly ameliorated. This is an interesting issue with clinical relevance. The methodology is well suited for this study. The authors used immunohistochemistry, pharmacology and behavioral analysis. However, a few issues need to be addressed.

1) Page 7. Section 2.15. I assume that these parametric tests were applied after verifying that the data has a normal distribution by the data normality tests, for example by Kolmogorov–Smirnov normality test. Please, indicate in the text this issue.

2) Page 7. Lines 378-380. I understand that the amount of DNA in the cytoplasm increased because the mitochondria were impaired. But how do you explain that the DNA of the whole cell increases?

3) Page 10, Lines 480-490. You indicate that mitochondria DNA was released via the Bak/Bax pore. Thus, why didn't HTHB increase the amount of BAK?

4) Discussion. There are some topics that are not covered in this section and should be included. It is known that this way of inducing sleep deprivation generates a lot of stress for the animal. Although I find it difficult to eliminate this effect in your studies (a control group with only stress), it should be mentioned in the discussion that stress also induce a microglia response.

5) Lines 640-648. Their data show that HTHB improves sleep deprivation disorders but in the same way that other compounds do, such as HB or HT. What are the differences between HTHB and the other products? You indicate that HTHB exhibited better behavioral performance however, this difference is not clear in Figure 1.

Reviewer 3 Report

Hu and colleagues performed this very interesting study assessing the effects of newly synthesized compound HTHB on neuroinflammation in a sleep-deprivation mouse model. Not surprisingly, they described a connection between oxidative stress and inflammation, with microglia cells playing an important role. Overall the study was well done, molecular mechanisms were described, but it has some limitations (see below).

The limitations of this study are:

1)      It is unclear to me why the authors focused on cerebral cortex (without specifying which part), as memory related parameters changed consistently. Molecular characterizations in the hippocampus should have been performed. Was something assessed to this regard? For example in Fig. 4 the hippocampus seems to be present in the images. It would be important to perform some (if not all) quantifications in this area, to provide a direct connection with the behavioral data. In addition, even within the cerebral cortex, different areas/layers should have been considered

2)      Branching analysis (sholl analysis) of microglia cells would have been very informative to assess the functionality of these cells

3)      It would have been interesting to see the effects of SD and HTHB on sociability.

Minor points:

-          The authors selected 72h as optimal time for the SD model. Did author ever tried later timepoints (for example 96h)?

-          When were the drugs administered? Before the 72h SD or just starting from day 0? Was a pre-treatment needed?

-          The authors could speculate a bit more about the possible benefits of their drug for SD conditions

-          TNF (and not TNF-a) is the right term.

Hu and colleagues performed this very interesting study assessing the effects of newly synthesized compound HTHB on neuroinflammation in a sleep-deprivation mouse model. Not surprisingly, they described a connection between oxidative stress and inflammation, with microglia cells playing an important role.

Overall the study was well done, but it has some limitations (listed below)

The limitations of this study are:

1)      It is unclear to me why the authors focused on cerebral cortex (without specifying which part), as memory related parameters changed consistently. Molecular characterizations in the hippocampus should have been performed. Was something assessed to this regard? For example in Fig. 4 the hippocampus seems to be present in the images. It would be important to perform some (if not all) quantifications in this area, to provide a direct connection with the behavioral data. In addition, even within the cerebral cortex, different areas/layers should have been considered

2)      Branching analysis (sholl analysis) of microglia cells would have been very informative to assess the functionality of these cells

3)      It would have been interesting to see the effects of SD and HTHB on sociability.

Minor points:

-          The authors selected 72h as optimal time for the SD model. Did author ever tried later timepoints (for example 96h)?

-          When were the drugs administered? Before the 72h SD or just starting from day 0? Was a pre-treatment needed?

-          The authors could speculate a bit more about the possible benefits of their drug for SD conditions

-          TNF (and not TNF-a) is the right term.

Reviewer 4 Report

Sleep deprivation is one of the major health concerns in todays world and significantly associates with a multitude of abnormal brain functions in humans. In this study by Hu et al., the authors investigated the underlying mechanisms of sleep deprivation-associated mitochondrial stress and genome instability that further triggers STING-mediated neuroinflammation in a rat model. To attenuate the oxidative stress on mitochondria, the authors applied an antioxidant compound called hydroxytyrosol butyrate that has been known to improve neurophysiological deficits in several other studies. The study has some serious drawbacks that demerit the implication of the findings in a broader sense.

The major comments include:

1) The chosen animal age may not be proper to investigate sleep-associated neuroinflammation, because aging plays a key role in modulating the oxidative stress in brain cells.

2) the key highlight of this study was mitochondrial pathology and its remediation. However, none of the experiments were performed in a purified mitochondrial extract.

1) Line 74 : "hydroxytyrosol butyrate (HTHB)" should be abbreviated as HTB. Not sure, what this extra "H" stands for.

2) Line 75: the cited Reference 35 could be tracked and/or retrieved from any database. Please provide further details and doi for this article. 

3) Line 82: The sentence has grammatical errors.

4) Please provide dose optimization and toxicity effects of HTHB in rats.

5) Fig. 2M requires Stat Quant.

6)  Fig. 3A: Iba1 should be costained with NeuN. Please include that result.

7) Fig. 3G: Why does the control show strong Iba1 signal?

8) Fig. 3E: Should be repeated with mitochondrial and cytosolic extracts not the whole cell lysates.

9) Fig. 4A: In the control group, NF-kB signal intensity is high. Were these rats stressed by any means?

10: Figs 4-6: Need to repeat with mitochondrial fractions.

Round 2

Reviewer 3 Report

The authors addressed all my points

Nothing to add

Author Response

Major comments: The authors addressed all my points.

Detail comments: Nothing to add.

Response: Thank you very much for your positive feedback. Your comments were crucial in improving the quality of our manuscript!

Reviewer 4 Report

The authors have improved the manuscript by adding additional information and supportive evidence. However, there are two major concerns that the authors should correct Fig 6E Y-axis label replacing "DNA expression" with "DNA levels", since it's the quantitation of amounts of released DNA in the cytosol not the gene expression before the final acceptance of the article.

Also, the authors should include the study limitations in the Discussion sections, as they did not perform any experiments with purified mitochondria.

Same as described in the previous section.
